# A Metric Learning-Based Univariate Time Series Classification Method

**Kuiyong Song [1,2], Nianbin Wang [1] and Hongbin Wang [1,\*]**

[1]   College of Computer Science and Technology, Harbin Engineering University, Harbin 150000, China;
    songkuiyong@hrbeu.edu.cn (K.S.); wangnianbin@hrbeu.edu.cn (N.W.)

[2]   Department of Information Engineering, Hulunbuir Vocational Technical College, HulunBuir 021000, China

\*   Correspondence: wanghongbin@hrbeu.edu.cn

**Abstract:** High-dimensional time series classification is a serious problem. A similarity measure based on distance is one of the methods for time series classification. This paper proposes a metric learning-based univariate time series classification method (ML-UTSC), which uses a Mahalanobis matrix on metric learning to calculate the local distance between multivariate time series and combines Dynamic Time Warping(DTW) and the nearest neighbor classification to achieve the final classification. In this method, the features of the univariate time series are presented as multivariate time series data with a mean value, variance, and slope. Next, a three-dimensional Mahalanobis matrix is obtained based on metric learning in the data. The time series is divided into segments of equal intervals to enable the Mahalanobis matrix to more accurately describe the features of the time series data. Compared with the most effective measurement method, the related experimental results show that our proposed algorithm has a lower classification error rate in most of the test datasets.

**Keywords:** Mahalanobis; metric learning; multivariable; time series; univariate

## 1. Introduction

Time series data are widely used in the real world, such as in the stock market [1], medical diagnosis [2], sensor detection [3], and marine biology [4]. With the deepening of studies on machine learning and data mining, time series is becoming a popular research field. Due to the high dimensionality and noise of time series data, in general, before analyzing the time series, dimension reduction and denoising of time series are very necessary. There are many common methods to reduce dimensionality and remove noise such as discrete wavelet transform (DWT) [5], discrete Fourier transform (DFT) [6], singular value decomposition (SVD) [7], piecewise aggregate approximation (PAA) [8], piecewise linear representation (PLR) [9], and symbolic aggregate approximation (SAX) [10].

Distance-based time series classification algorithms, such as the k-nearest neighbors (k-NN) [11] and support vector machines (SVM) [12], depend on the similarity measure of time series. The measure methods commonly used for time series include Euclidean distance, Mahalanobis distance [13], and DTW distance [14–16]. Euclidean distance is the most common method of calculating the point-to-point distance and is highly efficient and easy to calculate. However, the disadvantage is that it requires a series of equal lengths and intervals. Different from the Euclidean distance, DTW can calculate the distance between the series with different intervals. DTW seeks the shortest path between the series distances and calculates the similarity by stretching or shrinking the time series. It can also incorporate series distortion or translation. However, the complexity of DTW is high, and the efficiency is low if high-dimensional sequences are calculated.

Mahalanobis distance is used to measure multivariable time series data. The traditional Mahalanobis matrix, based on covariance matrix inversion, is generally used to reflect the internal

aggregation relations of data. However, in most classifications, it is not suitable for using a distance metric because it only reflects the internal aggregation, whereas it is more important to establish the relation between sample attributes and classification. In [17–19], metric learning is used to solve measurement problems in multivariate time series similarity, and a better result is obtained. Distance metric learning is used to obtain a Mahalanobis matrix that can reflect distances between data effectively by learning from training samples. In a new feature space, distributions of intraclass samples are closer, while interclass samples are spread further. Common distance metric learning methods include probabilistic global distance metric learning (PGDM) [20], large margin nearest neighbor learning (LMNN) [21], and information-theoretic metric learning (ITML) [22].

In recent years, many univariable time series classification methods have been proposed. The SAX and SAX_TD [23] algorithms are based on feature representations. In SAX and SAX_TD, intervals of equal probability are segmented based on PAA, and each of the intervals is represented with symbols to transform the time series into a symbol string. To some extent, SAX can compress the data length and reduce the dimensions. However, due to the adoption of PAA, the peak information is lost, resulting in low accuracy. Based on DTW deformation, LCSS [24] and EDR [25], similar to DTW, these have the problem of high-time complexity. Ye and Keogh [26] and Grabocka et al. [27] presented shapelet-based algorithms which require high-time complexity for generating a large number of shapelet candidates. We can conclude that there are three main problems with the above algorithms:

- How to treat higher dimensional time series data.
- How to find a suitable distance measure method to improve classification accuracy.
- How to compare unequal time series.

To address these problems. A novel method, ML-UTSC, is proposed in this paper to classify univariate time series data. First, PLR was adopted to reduce the dimensions of the time series. Compared to PAA, the series tendency and peak information were maintained. Second, the mean value, variance, and slope of the fitting lines were calculated to form a triple. The univariate time series was transformed into a multivariate time series, and metric learning was used to learn the Mahalanobis matrix. Finally, the combination of the Mahalanobis matrix with DTW is used to calculate the multivariate time series distance.

In this work, we make three main contributions. First, the problem of classifying univariate time series data by metric learning is proposed for the first time. Second, to ensure the consistency of univariate feature representation, and that the time series is divided equally. Third, the experimental results show that the Mahalanobis matrix obtained by metric learning has a better classification effect.

The rest of the article is organized as follows. The related background knowledge is introduced in the second part. The ML-UTSC algorithm is described in the third part. The experimental comparison results and analysis are given in the fourth part. The fifth part concludes the manuscript.

## 2. Background

### 2.1. Dimension Reduction

PLR is a method to represent piecewise linear fitting. It can compress a time series of length $n$ into $k$ straight lines ($k < n$), which may the make data storage and calculation more efficient. Least squares linear fitting is one of the most effective PLR methods. The linear regression is described using the following Equation:

$$\hat{y}_i = \hat{\beta}_0 + \hat{\beta}_1 x_i \tag{1}$$

For $n$ points with equal intervals, $(x_i, y_i)$, $i = 1, 2, \ldots n$. $x_i$ and $y_i$ are the abscissa and ordinate values of a point, respectively. $\hat{y}_i$ is the fitting value of point $(x_i, y_i)$. $\hat{\beta}_0$ is an intercept, and $\hat{\beta}_1$ is the

corresponding slope. The error is only related to $\hat{y}_i$ and $y_i$. The fitting error of the least-squares fitting method [8] is shown in the following Equation:

$$\sum (y_i - \hat{y}_i)^2 = \sum (y_i - \hat{\beta}_0 - \hat{\beta}_1 x_i)^2 \tag{2}$$

Defining $Q$ equal to (2), we can calculate the partial derivative of $Q$ corresponding to $\hat{\beta}_0$ and $\hat{\beta}_1$, according to the mean value theorem. Then, when it is set to 0,

$$\begin{cases} \left.\frac{\partial Q}{\partial \beta_0}\right|_{\beta_0 = \hat{\beta}_0} = -2\sum_{i=1}^{n} (y_i - \hat{\beta}_0 - \hat{\beta}_1 x_i) = 0 \\ \left.\frac{\partial Q}{\partial \beta_1}\right|_{\beta_1 = \hat{\beta}_1} = -2\sum_{i=1}^{n} x_i(y_i - \hat{\beta}_0 - \hat{\beta}_1 x_i) = 0 \end{cases} \tag{3}$$

then, the Equation in (3) yields a linear system easy to solve.

### 2.2. Metric Learning

In studies on metric learning [17], Mahalanobis distance is not defined as the inversion of covariance but should be obtained by metric learning. If there are two multivariate sequences $x_i$ and $x_j$, a positive semidefinite matrix $M$ is given called the Mahalanobis matrix. The Mahalanobis distance can be formalized as follows:

$$D_M(x_i, x_j) = (x_i - x_j)^T M (x_i - x_j) \tag{4}$$

$D_M(x_i, x_j)$ is the Mahalanobis distance between $x_i$ and $x_j$. The distance metric learning obtains a metric matrix that reflects the distances between the data by learning a given training sample set. The goal of metric learning is to determine the matrix $M$. To ensure that the distance is nonnegative and to satisfy the triangle inequality, $M$ should be a positive definite (semidefinite) symmetric matrix. That is, there is an orthogonal basis $P$ with the property $M = PP^T$.

PGDM is a typical algorithm that transforms metric learning into a constrained convex optimization problem. Taking the chosen pair constraints as a constraint condition of the training sample, the main idea is to minimize the distance between intraclass samples when the constrained distance between interclass sample pairs is greater than a certain value. The optimized model is as follows:

$$\min_{M} \sum_{(x_i, x_j) \in S} \| x_i - x_j \|_M^2$$
$$s.t. \sum_{(x_i, x_j) \in D} \| x_i - x_j \|_M \geq 1, M \geq 0 \tag{5}$$

If $M$ is found using the Mahalanobis matrix, then, for any intraclass samples $x_i$ and $x_j$, the squared sum of the distances is minimized. Additionally, the constrained condition is that the distance between the interclass samples $x_i$ and $x_j$ is greater than 1 and $M$ is a positive semidefinite. The PGDM loss function is then

$$g(M) = g(M_{11}, \cdots, M_{nn}) = \sum_{(x_i, x_j) \in S} \|x_i - x_j\|_M^2 - \log\left(\sum_{(x_i, x_j) \in D} \|x_i - x_j\|_M\right) \tag{6}$$

The loss function is equivalent to the optimized model in (5) when they solve a convex optimization problem, which can be solved with methods, including Newton and quasi-Newton.

### 2.3. Dynamic Time Warping

For two time series $q = \{q_1, q_2, \dots, q_m\}$ and $c = \{c_1, c_2, \dots, c_n\}$, a matrix $D$ is constructed where $d_{ij}$ is the Euclidean distance between $q_i$ and $c_j$. DTW finds an optimal path $w = \{w_1, w_2, \dots, w_K\}$ where $w_k$ is the location of the corresponding elements and $w_k = (i,j)$, $i \in [1:m]$, $j \in [1:n]$, $k \in [1:K]$, so DTW of $q$ and $c$ is,

$$\text{DTW}(q,c) = \sqrt{\sum_{k=1, w_k=(i,j)}^{K} \left(q_i - c_j\right)^2} \tag{7}$$

The optimal path $w$ can be obtained through dynamic programming with the distance matrix $D$:

$$R(i,j) = d_{ij} + min\{R(i,j-1), R(i-1,j-1), R(i-1,j)\} \tag{8}$$

where $R(0,0) = 0$, $R(i,0) = R(0,j) = +\infty$. $R(m,n)$ is the minimum distance between the time series $q$ and $c$.

## 3. ML-UTSC

### 3.1. Least Squares Fitting

The univariate time series $x$ and $y$ are given as $x = \{(t_1, x_1)(t_2, x_2), \ldots, (t_m, x_m)\}$, $y = \{(t_1, y_1), (t_2, y_2, \ldots, (t_n, y_n)\}$, where $m$ and $n$ are the time series dimensions. To reduce the series dimensions, least-squares fitting is performed and the time series is divided into several segments. The bottom-up time series leas- squares fitting is divided into two steps. First, each point is taken as a basic unit and the adjacent points are combined, and then each segment is fit. If the fitting error is lower than the threshold *max_error*, combining of the time series continues until the error exceeds the threshold, and the combination stops when the fitting error (2) fits:

$$\sum \left(y_i - \hat{y}_i\right)^2 = \sum \left(y_i - \hat{\beta}_0 - \hat{\beta}_1 x_i\right)^2 \geq max\_error \tag{9}$$

The fitting results of some time-series data are shown in Figure 1. A group of data was selected in the 50 Words dataset, in which the length of the data was 270. Figure 1A is the original data set, and Figure 1B–D are the least-squares fitting diagrams with *max_error* rates of 0.1, 0.5, and 1, respectively.

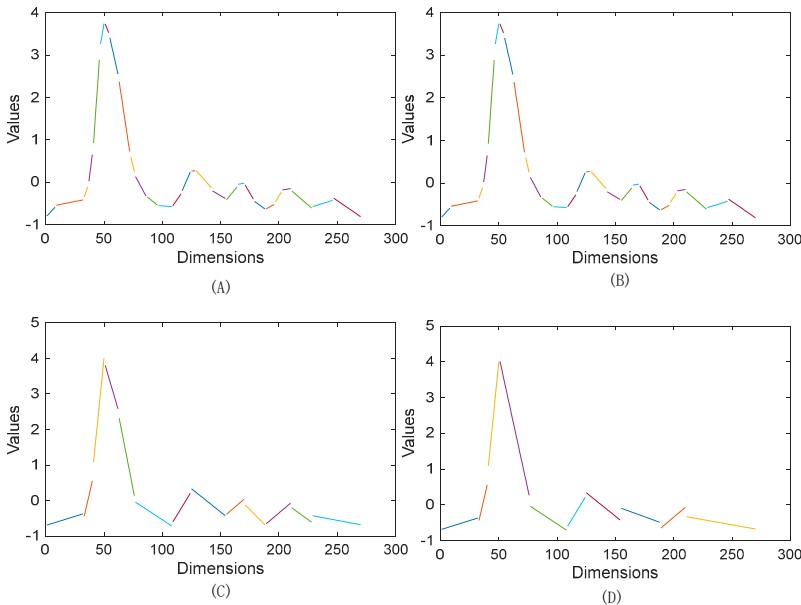

**Figure 1.** The least-squares fitting in different *max_error* on a 50 Words dataset. (**A**) the origin data; (**B**) *max_error* = 0.1; (**C**) *max_error* = 0.5; (**D**) *max_error* = 1.

It can be seen from these figures that as the max_error increases, the number of fitting segments decreases, and the figures run from smooth to rough. In terms of the accuracy of the feature representation, as the max_error increases, the number of segments decreases, and the dimension reduction rate increases, which makes the feature representation accuracy lower. The pseudocode for bottom-up time-series least-squares fitting is given in Algorithm 1.

---

**Algorithm 1: [TS] = Bottom_Up_Fitting(T,*max_error*)**

---

```
    1:   for i = 1:2:length(T)
 2:      TS = concat(TS, create_segment(T[i:i + 1]));
 3:   end;
 4:   for i = 1:length(TS) − 1
 5:      merge_cost(i) = calculate_error([merge(TS(i), TS(i + 1))]);
 6:   end;
 7:   while min(merge_cost)<max_error
 8:      index = min(merge_cost);
 9:      TS(index) = merge(TS(index), TS(index + 1));
10:      delete (TS(index + 1));
11:      merge_cost(index) = calculate_error(merge(TS(index);
12:      merge_cost(index − 1) = calculate_error(merge(TS(index − 1);
13:   end
```

---

### 3.2. Feature Representation

To reduce the dimensions and eliminate the influence of noise, least-squares fitting is used to linearly represent the time-series segments. Here, segments are further characterized by the mean value $E$, variance $V$, and slope values $S$. Thus, the triple $(E, V, S)$ was constructed to represent the time series segments. The triple matrix of time series x is

$$Tx = \begin{bmatrix} E_1 & E_2 & E_3 & \cdots & E_K \\ V_1 & V_2 & V_3 & \cdots & V_K \\ S_1 & S_2 & S_3 & \cdots & S_K \end{bmatrix} \tag{10}$$

where $K$ is the number of segments in the time series fitting. Therefore, the features of univariate time series data can be represented by three variables. However, a problem may occur when different line segments have the same mean and slope values. As shown in Figure 2A, there are three parallel segments, $l_1$, $l_2$, and $l_3$, with lengths of 5, 10, and 15, respectively. Their mean values and slope values are the same, while the variances are different. If the triple is used to calculate the distance between $l_1$ and $l_2$, the mean value and slope values have no meaning. However, it does not reflect their properties because the lengths of $l_1$ and $l_2$ are different. To reflect the feature of lines more accurately, dividing segments into equal intervals (weights) is the most feasible.

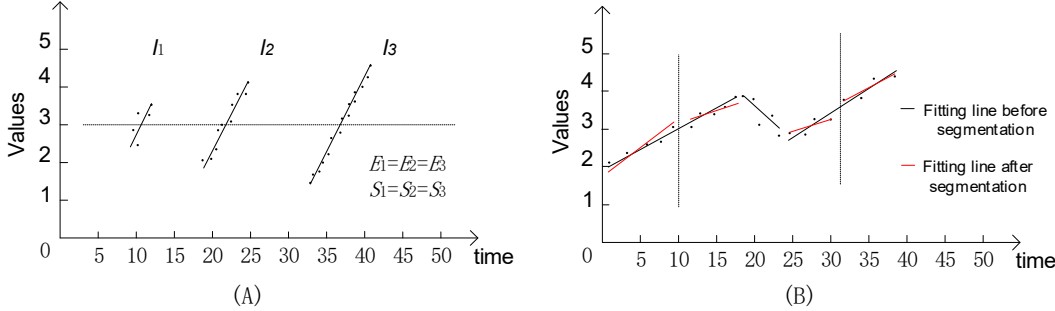

**Figure 2.** The characteristic analysis and equal interval segmentation of the line. (**A**) The same mean and slope values for $l_1$, $l_2$, and $l_3$; (**B**) Equal interval segmentation of the line.

Figure 2B shows that the time series is three black segments after least-squares fitting in which the lengths are 10, 5, and 8. Stipulating that the interval distance d is 5, the results of equal intervals are red segments. After equal interval segmentation, the mean value, variance, and slope values all change, and the fitting must be calculated again. The first segment is divided into two equal parts, and the second is not divided, while the third is divided into two equal parts. It can be seen from Figure 2B

that the time lengths of the red divided and refitted segments are almost the same, and the weights are also almost identical.

Using the 50 Words dataset and stipulating that *max_error* is 0.1, the least-squares fitting results are shown in Figure 1B. It can be seen from the figure that the time intervals of the segments are different, ranging from 4 to 24. With an interval distance *d* of 5, the results using equal intervals are shown in Figure 3. In addition, the segment time lengths are all approximately 5, with little difference in value. Compared with Figure 1B, the entire series segment is smoother.

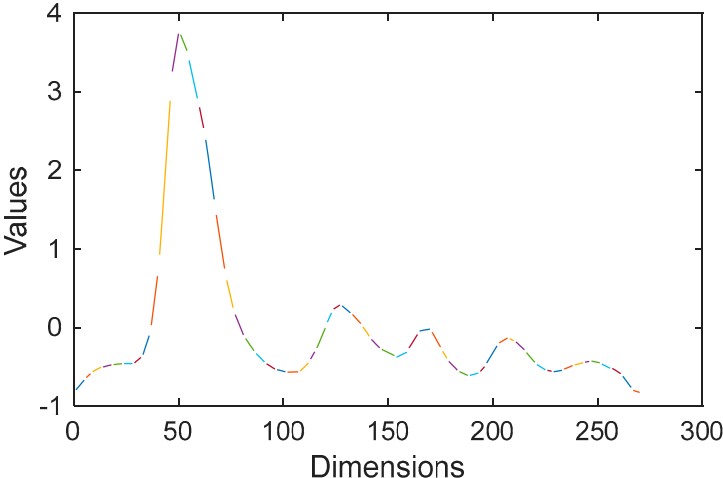

**Figure 3.** The fitting line of the data after segmenting with *d* = 5, *max_error* = 0.1.

It cannot be guaranteed that each of the segments is the same after segmentation. For instance, the length of the third segment after segmentation is 4 in Figure 1B. However, the homogeneity of the segments can be guaranteed. In addition, the time series is represented as a matrix of triples after equal interval segmentation.

### 3.3. Metric Learning

DTW is often used to calculate the univariable time series distance. In [28], DTW was extended to a multivariable time series, and the Euclidean distance was used for the local distance. The Euclidean distance considers each variable without considering the relationship between variables and is affected by noise and irregularity. In [19], DTW based on the Mahalanobis distance was used to calculate the multivariable distance for the first time. The Mahalanobis distance assigns different weights for different variables, and the relationships between variables are considered.

3.3.1. Calculate Multivariate Local Distance

As described above, the features of the time series are represented as a matrix of triple ($E_k$, $V_k$, $S_k$). In a triple matrix, each point is a vector. Therefore, the local distance between the two triples is the distance between two vectors. The basic structure is shown in Figure 4, where *Tx* and *Ty* are two matrices of triple, the middle part of Figure 4 is the optimal path of DTW, and the local distance is calculated by the Mahalanobis matrix. In this paper, the Mahalanobis matrix based on measurement learning is used as the local distance, and the distance of the multivariable sequence is calculated by combining DTW.

As shown in (4), if there are triple matrices *Tx* and *Ty*, the local distance is calculated as:

$$D_M\left(Tx^i, Ty^j\right) = \left(Tx^i - Ty^j\right)^T M\left(Tx^i - Ty^j\right)$$
$$1 \leq i \leq m, 1 \leq j \leq n$$

(11)

where $Tx^i$ and $Ty^j$ are the *i*th and *j*th columns of matrix $Tx$ and $Ty$, respectively. Combining (8) and (11) gives

$$R_M(i,j) = d_M(Tx^i, Ty^j) + min\{R_M(i,j-1), R_M(i-1,j-1), R_M(i-1,j)\} \tag{12}$$

where *m* is the column number of $Tx$ and *n* is that of $Ty$. The difference from formula (4) is that the Mahalanob distance is used instead of the Euclidean distance. Thus, DTW($Tx$,$Ty$) is equal to $R_M(m,n)$.

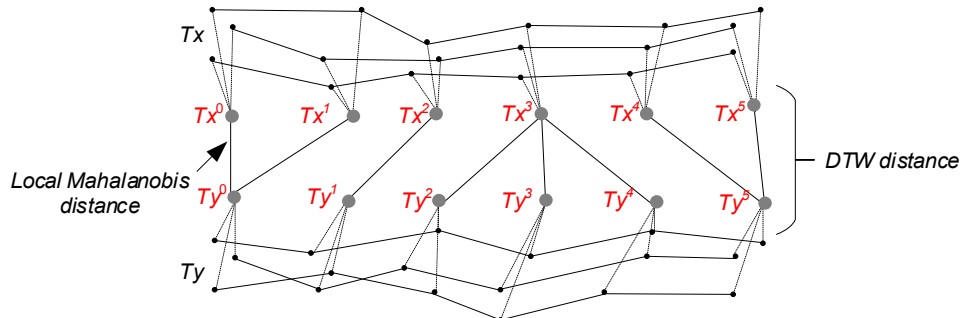

**Figure 4.** The optimal path of the DTW and the local Mahalanobis distance.

### 3.3.2. Learning A Mahalanobis Matrix

In (12), a good Mahalanobis matrix *M* was able to accurately reflect the multivariate measurement in certain spaces [17]. To obtain a better Mahalanobis matrix, PGDM was selected in this paper. However, PGDM is able to learn with unordered data but fails to process time-series data. To learn a "good" Mahalanobis matrix, PGDM and DTW were combined as a learning algorithm for the time-series data.

First, the DTW is a dynamic programming process that causes the loss function to be nondifferentiable. Therefore, metric learning should transform the DTW into general paths. An optimized path $w = \{w_1, w_2, \dots, w_K\}$, where $w_k = (w_x(k), w_y(k))$, is found with the DTW method and the extracted general path is:

$$\begin{cases} \overline{X}^k = Tx^{(w_x(k))} \\ \overline{Y}^k = Ty^{(w_y(k))} \end{cases} k = 1, 2, \cdots K \tag{13}$$

Based on this path, the DTW distance is transformed into the general path distance:

$$D_M(Tx, Ty) = \sum_{k=1}^{K} (\overline{X}^k - \overline{Y}^k)^T M (\overline{X}^k - \overline{Y}^k) \tag{14}$$

Then, the PGDM optimized loss function is updated by (6) and (14):

$$g(M) = \sum_{(Tx,Ty)\in S} D_M^2(Tx, Ty) - \log(\sum_{(Tx,Ty)\in D} D_M(Tx, Ty)) \tag{15}$$

Combining (14) and (15) gives:

$$\begin{aligned} g(M) = \quad & \sum_{(Tx,Ty)\in S} (\sum_{k=1}^{K} (\overline{X}^k - \overline{Y}^k)^T M (\overline{X}^k - \overline{Y}^k))^2 \\ & - \log(\sum_{(Tx,Ty)\in D} \sum_{k=1}^{K} (\overline{X}^k - \overline{Y}^k)^T M (\overline{X}^k - \overline{Y}^k)) \end{aligned} \tag{16}$$

Finally, the transformed loss function is differentiable and can be optimized with Newton's method or the conjugate gradient method. In [18], a greedy strategy that considered the minimization process as an iterative two-step optimization process was proposed. For this algorithm, first, after fixing the Mahalanobis matrix *M*, the optimized path between two multivariates is sought. Then, the gradient

method is used to minimize the loss function. Theoretically, this method can ensure convergence, but not global convergence because the loss function is nonconvex. In practice, even though it may reach a local optimization, the classification performance is usually good.

The time cost of ML-UTSC includes two parts. The first part is data preprocessing and triple matrix generation. The second is the optimization with PGDM; usually the classification performs well. In the data preprocessing step, the bottom-up least-squares fitting strategy was adopted with the time complexity $O(ln)$, where $n$ is the average length of the time series, and $l$ is the number of segments. Additionally, in the PGDM optimization, the approximate complexity is $O(n^2)$, where $n$ is the average length of the time series. Therefore, the time complexity of the ML-UTSC algorithm is $O(n^2)$.

## 4. Experimental Confirmation

To verify the validity of ML-UTSC, time-series datasets were selected from the UCR Time Series Classification Archive to compare the error rate, dimensionality reduction, and time efficiency of the algorithm under different parameters. It can be found at http://www.cs.ucr.edu/~{}eamonn/time_series_data/. All the tests in this paper were performed in the MATLAB 2016a environment and on the same computer with an Intel Core i5-4590, 3.3 Ghz, 8 GB memory, and WINDOWS 10.

### 4.1. Data Set

A total of 20 representative time-series datasets were selected from the UCR Time Series repository, as shown in Table 1, which includes the dataset name, number of categories, number of training sets, number of test sets, length, and type of time series. The number of dataset categories ranged from 2 to 50, the number of training sets ranged from 24 to 1000, the number of test sets ranged from 30 to 900, and the time series length ranged from 60 to 6174. In addition, the dataset type included synthetic, real (recorded from some processes), and shape (extracted by processing some shapes) [23].

**Table 1.** Information on the datasets.

| No. | Dataset Name | Classes | Size of Training Set | Size of Testing Set | Length of Time Series |
|-----|-------------|---------|---------------------|--------------------|----------------------|
| 1 | Synthetic Control | 6 | 300 | 300 | 60 |
| 2 | Gun-Point | 2 | 50 | 150 | 150 |
| 3 | CBF | 3 | 30 | 900 | 128 |
| 4 | Face (all) | 14 | 560 | 1690 | 131 |
| 5 | OSU Leaf | 6 | 200 | 242 | 427 |
| 6 | Swedish Leaf | 15 | 500 | 625 | 128 |
| 7 | 50Words | 50 | 450 | 455 | 270 |
| 8 | Trace | 4 | 100 | 100 | 275 |
| 9 | Two Patterns | 4 | 1000 | 4000 | 128 |
| 10 | Water | 2 | 1000 | 6174 | 152 |
| 11 | Face (four) | 4 | 24 | 88 | 350 |
| 12 | Lightning-2 | 2 | 60 | 61 | 637 |
| 13 | Lightning-7 | 7 | 70 | 73 | 319 |
| 14 | ECG | 2 | 100 | 100 | 96 |
| 15 | Adiac | 37 | 390 | 391 | 176 |
| 16 | Yoga | 2 | 300 | 3000 | 426 |
| 17 | Fish | 7 | 175 | 175 | 463 |
| 18 | Beef | 5 | 30 | 30 | 470 |
| 19 | Coffee | 2 | 28 | 28 | 286 |
| 20 | Olive Oil | 4 | 30 | 30 | 570 |

### 4.2. Comparison Methods and Parameter Setting

In order to verify the effectiveness of ML-UTSC, three different methods were selected for comparison, namely Euclidean Distance(EUC), SAX_TD, and DTW. Due to the compression of data in

this paper, SAX_TD, a similar method, was selected for comparison. SAX_TD accounts for the trend information and achieves higher classification accuracy. DTW is a classic elastic measurement method that can measure unequal-length time series with high scalability and accuracy. The experiments [16] show that DTW is still one of the methods with the highest accuracy of time series classification. In addition, to verify the effect of equidistant segmentation on the classification error rate of the ML-UTSC algorithm, the ML-UTSC-B was marked as ML-UTSC without equidistant segmentation. In the ML-UTSC algorithm, the least-squares fitting threshold is the *max_error* rate, and the equidistant segmentation threshold is *d*. Additionally, in ML-UTSC_B, only the *max_error* rate is needed.

To obtain better accuracy for SAX_TD and ML-UTSC, we set different parameters for testing, and the highest accuracy and corresponding parameters were recorded. For a given time series with length *n*, SAX_TD takes the argument *w* from 2 to *n*/2, multiplying by 2 at a time, and the argument *α* value is set from 3 to 10 [23]. ML-UTSC takes the values of the *max_error* rate to be 0.1, 0.5, 1, 1.5, and 2, while the values of *d* were 5, 10, 15, 20, and 25. The dimensionality reduction rate was equal to the number of reduced data points divided by the number of source data points. In the experimental analysis, it was found that such parameters were able to meet the dimensionality reduction range criteria.

### 4.3. Classification Results Analysis

The results of the five methods on the 20 datasets are listed in Table 2. In the parentheses of SAX-TD and ML-UTSC are the parameters used to obtain the value reported. The minimum error rate in each row is shown in bold, and in the 20 datasets, there were 12 minimum values in ML-UTSC, five in DTW, and two in SAX-TD. However, multiple values with the same minimum values are not shown in bold; for instance, there are four methods that obtain the minimum value in the 19$^{th}$ dataset. By comparing the number of minimum values, it was found that ML-UTSC has a lower error rate for most of the datasets, and the value did not differ from the minimum even if the minimum error rate was not obtained. Additionally, the average error rate of the ML-UTSC was only 0.07 higher than the lowest average error rate in the other eight datasets with no minimum error rate. From the error rates of ML-UTSC and ML-UTSC-B, it can be clearly seen that the error rate of ML-UTSC was lower than the error rate before segmentation.

**Table 2.** Comparision of parameters for the five algorithms used.

| No. | Dataset Name | EUC *Error* | SAX-TD *Error (w,α)* | DTW *Error* | MLUTSC_B *Error* | ML-UTSC *Error(max_error, d)* |
|---|---|---|---|---|---|---|
| 1 | Synthetic Control | 0.120 | 0.077 (8,10) | **0.017** | 0.152 | 0.053 (1,10) |
| 2 | Gun-Point | 0.087 | 0.073 (4,3) | 0.093 | 0.046 | **0.026** (0.1,5) |
| 3 | CBF | 0.148 | 0.088 (4,10) | **0.004** | 0.014 | 0.011 (1,5) |
| 4 | Face (all) | 0.286 | 0.215 (16,8) | **0.192** | 0.244 | 0.210 (0.5,10) |
| 5 | OSU Leaf | 0.483 | 0.446 (32,7) | 0.409 | 0.415 | **0.388** (0.5,10) |
| 6 | Swedish Leaf | 0.211 | 0.213 (16,7) | 0.208 | 0.255 | **0.188** (0.1,5) |
| 7 | 50Words | 0.369 | 0.338 (128,9) | 0.310 | 0.323 | **0.284** (0.5,5) |
| 8 | Trace | 0.240 | 0.210 (128,3) | **0.010** | 0.152 | 0.084 (0.1,10) |
| 9 | Two Patterns | 0.093 | 0.071 (16,10) | **0.002** | 0.086 | 0.023 (0.5,5) |
| 10 | Water | 0.005 | 0.004 (32,8) | 0.020 | 0.005 | **0.001** (0.1,10) |
| 11 | Face (four) | 0.216 | 0.181 (32,9) | 0.170 | 0.352 | **0.147** (0.1,5) |
| 12 | Lightning-2 | 0.246 | 0.229 (8,9) | 0.131 | 0.163 | **0.114** (0.5,10) |
| 13 | Lightning-7 | 0.425 | 0.329 (16,10) | 0.274 | 0.312 | **0.246** (1,10) |
| 14 | ECG | 0.120 | **0.092** (16,5) | 0.230 | 0.252 | 0.220 (0.1,5) |
| 15 | Adiac | 0.389 | 0.273 (32,9) | 0.396 | 0.351 | **0.224** (0.5,10) |
| 16 | Yoga | 0.170 | 0.179(128,10) | 0.164 | 0.155 | **0.102** (1,5) |
| 17 | Fish | 0.217 | 0.154 (64,9) | 0.177 | 0.213 | **0.151** (1,10) |
| 18 | Beef | 0.467 | 0.200 (64,9) | 0.367 | 0.421 | **0.266** (1,5) |
| 19 | Coffee | 0.250 | 0.000 (8,3) | 0.000 | 0.000 | 0.000 (0.1,5) |
| 20 | Olive Oil | 0.133 | **0.067** (64,3) | 0.167 | 0.267 | 0.233 (0.1,5) |

However, it was observed from Table 2 that in six datasets with a length less than 150, including Synthetic Control, ECG, CBF, Face (all), Two Patterns, Swedish Leaf, on the first five datasets, ML-UTSC was not competitive, and the Swedish Leaf was not significantly different from the other three algorithms. That is, the Mahalanobis matrix learned by shorter sequences was insufficient to reflect the internal relations of the new feature space, which is the deficiency of ML-UTSC.

To further verify the test results, the ML-UTSC and other methods were compared by a sign test, and it was found that a smaller significance level of the results shows an obvious difference. In Table 3, $n_+$, $n_-$, and $n_0$ are used to represent the number of ML-UTSC' error rates, which is less than, above, or equal to that of other methods.

**Table 3.** Comparison of *p*-values by different methods.

| Methods | $n_+$ | $n_-$ | $n_=$ | *p-Value* |
|---|---|---|---|---|
| ML-UTSC VS. EUC | 18 | 2 | 0 | $p < 0.01$ |
| ML-UTSC VS. SAX-TD | 16 | 3 | 1 | $0.01 < p < 0.05$ |
| ML-UTSC VS. DTW | 13 | 6 | 1 | $p = 0.167$ |
| ML-UTSC VS. ML-UTSC_B | 19 | 0 | 1 | $p < 0.01$ |

In addition, the *p*-value is notable when ML-UTSC is compared with other methods. The *p*-value in Table 3 indicates that ML-UTSC is particularly significant when compared with EUC. ML-UTSC is significant when compared with SAX-TD, and ML-UTSC is, on average, significant when compared with DTW.

The minimum error rate of ML-UTSC in Table 2 was obtained with different parameters. To test the effect of the *max_errormax_error* parameter and *d* on the classification error rate, three datasets, including Face (four), Lightning-2, and Fish, were selected. The test results are shown in Figure 5.

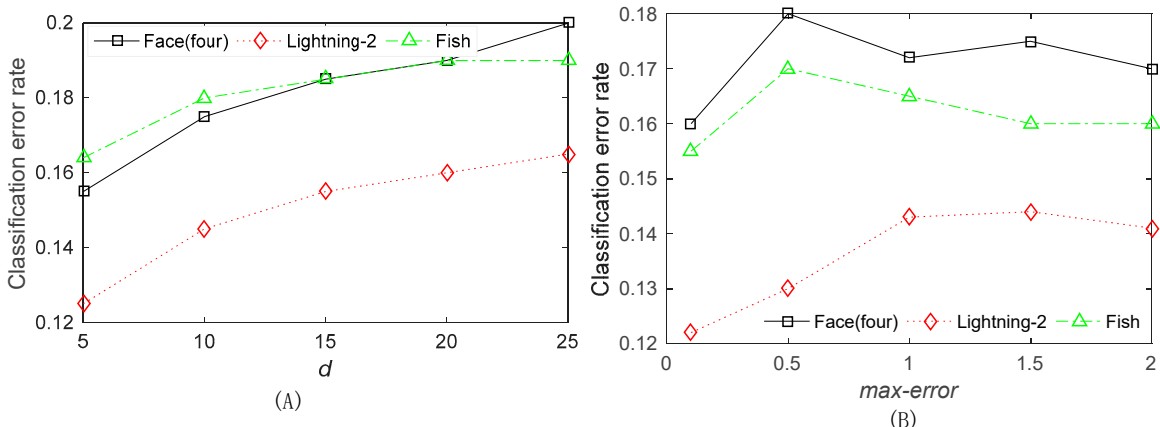

**Figure 5.** The trends comparison of the classification error rate under different parameters. (**A**) *max_error* is fixed and *d* is variable; (**B**) *d* is fixed and *max_error* is variable.

To test the effect of *d*, as shown in Figure 5A, the *max_error* is set as 0.5 initially, the values of *d* are 5, 10, 15, 20, 25, and the vertical axis shows the classification error rate. It can be seen that as the value of *d* increased, the error rates of the three datasets also increased slowly, which indicates that a smaller segmentation would make the error rate lower. To test the effect of the *max_error* rate on the classification error rate, as reported in Figure 5B, the value of *d* is set as 10 initially, the values of the *max_error* are 0.1, 0.5, 1, 1.5, and 2, and the vertical axis is the classification error rate. The trend showed that as the value of the *max_error* rate increases, the error rates of the three datasets slowly increase. However, when the value of the *max_error* rate is more than 1, the overall error rates of the three datasets began to decrease.

According to the analysis in Figure 5, the initial value of the *max_error* rate was smaller, and the segment length was very small, so the value of *d* had no effect. Therefore, the classification error rate was low. As the value of the *max_error* rate increased, the error rate also increased. In addition, when the segment length reached a certain level, the error rate could be reduced by reducing the segment length with equidistance segmentation.

The scatter diagram is an effective visualization method to compare the error rate. In Figure 6, four scatter matrices are plotted, and the values of the axes are the error rates of the two methods. The diagonal divides the matrix into two regions. The region with more points indicates that the method achieved lower error rates in most of the datasets. In addition, the farther the distance to the diagonal is, the larger the difference.

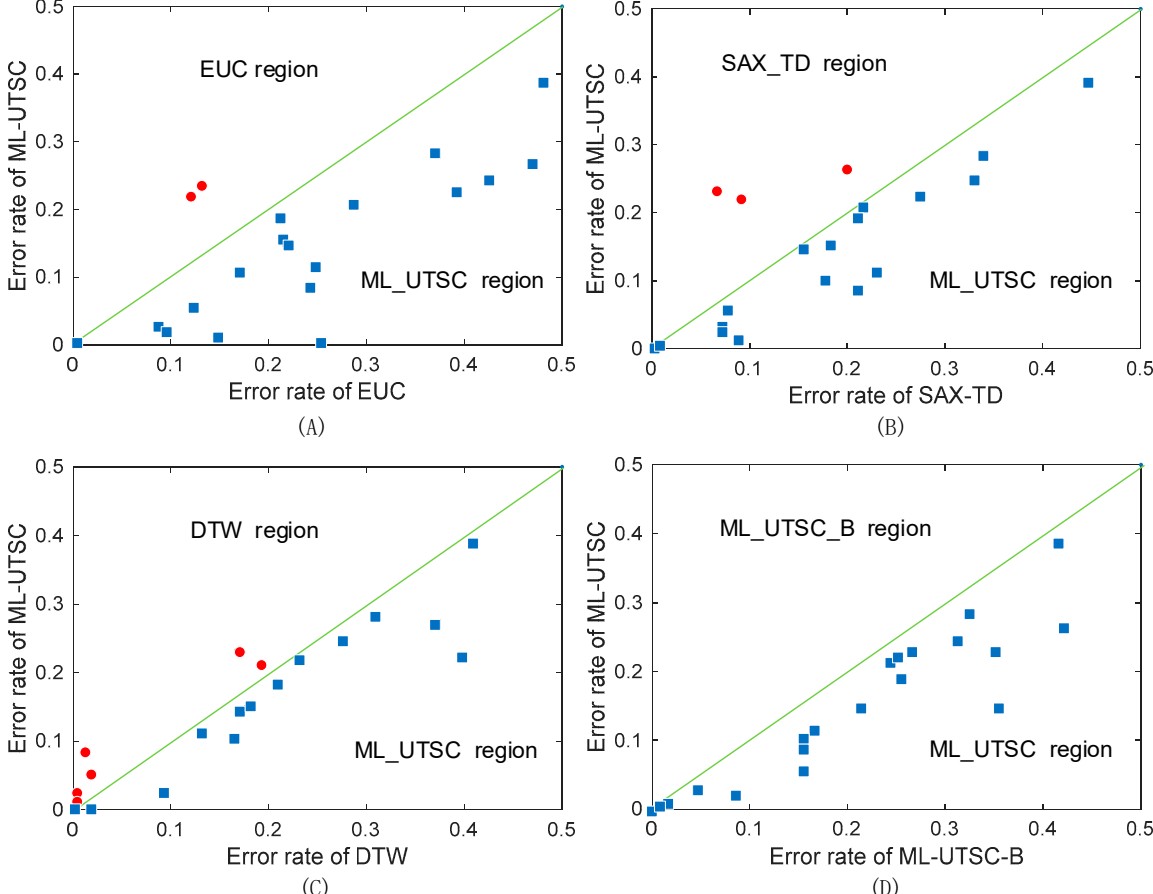

**Figure 6.** Scatter comparison of the five methods. (**A**) EUC VS. ML_UTSC; (**B**) SAX_TD VS.ML_UTSC; (**C**)DTW VS. ML_UTSC; (**D**) ML_UTSC_B VS. ML_UTSC.

Figure 6A compares EUC and ML-UTSC. It can be seen from the figure that most of the blue rectangular points are on the ML-UTSC region, and there are only two red points on the EUC region, In addition, most of the blue rectangular points are far away from the diagonal, which indicates that ML_UTSC is much better than EUC in most datasets. In Figure 6B,C, there are not too many red points, and there are many blue rectangular points around the diagonal in Figure 6C, which indicates that ML_UTSC is better than SAX_TD and DTW, and DTW is closest to ML_UTSC. In Figure 6D, there is no point in the ML-UTSC-B region, which indicates that the ML-UTSC after segmentation has lower error rates on all datasets.

### 4.4. Dimension Reduction and Time Efficiency

In the five test methods, the dimensionality of the data in SAX-TD, ML-UTSC-B, and ML-UTSC was reduced. In SAX-TD, the size of the data was reduced, and if the number of segments was $w$, the dimensionality reduction rate was $(2w + 1)/n$. Data in the ML-UTSC-B were reduced by the least-squares fitting with the reduction rate related to the threshold value *max_error*. In addition, the smaller the *max_error* value, the lower the reduction rate. For instance, when the value of *max_error* was 0.1, the reduction rate was generally 1/5 of the dataset. When the value of the *max_error* rate was 1, the reduction rate was generally 1/15 of the dataset. When the ML-UTSC was performed with equidistant segments based on ML-UTSC-B, the dimensionality reduction rate is determined by the *max_error* rate and $d$ together. Generally, if the value of the *max_error* rate was set smaller, $d$ would have less influence on the reduction rate. If the value of *max_error* was larger, the fitting segment would be larger, and $d$ would have a greater influence on the reduction rate. In Table 2, the *max_errors* in ML-UTSC_B and ML-UTSC are the same, which makes the comparison clearer. As shown in Figure 7, the dimensionality reductions of SAX-TD, ML-UTSC-B, and ML-UTSC that gave the lowest error rates are compared.

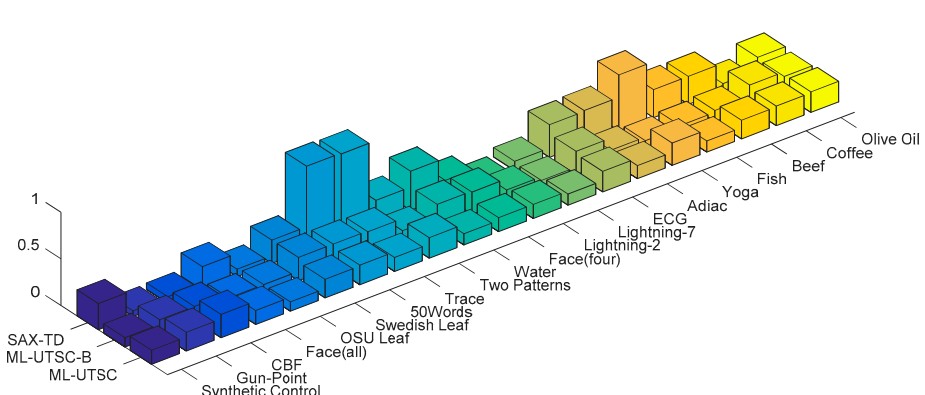

**Figure 7.** Comparison of dimension reduction for SAX-TD, ML-UTSC-B, and ML-UTSC.

It can be seen from Figure 7 that the straight squares of SAX-TD mostly have higher reduction rates than the other two methods. However, the rates are higher only in certain datasets. For instance, the reduction rate in Gun-Point, CBF, Lightning-2, and Lightning-7 was only approximately 1/10 of the dataset. However, the reduction rate on 50Words, Trace, and Yoga was significantly lower because the value of $w$ was 128 when the minimum error rate was obtained on the three datasets. Therefore, there was almost no reduction. Compared with SAX-TD, the reduction rates of ML-UTSC_B and ML-UTSC were higher in most of the datasets, and slightly lower in certain other datasets. Additionally, there was not much difference between ML-UTSC_B and ML-UTSC, and both have their own advantages.

Finally, the time efficiencies of EUC, SAX-TD, DTW, and ML-UTSC were compared, and the Synthetic Control, ECG, and CBF datasets were selected. The time efficiencies were compared under a minimum classification error rate. The total time included data preprocessing time and classification time, excluding the time in the metric learning Mahalanobis distance. The time taken by the four algorithms is shown in Figure 8.

It can be observed from Figure 8 that EUC required the least time, and DTW required the most time. This result can be confirmed by the analysis of time complexity. Without considering the time in the metric learning Mahalanobis matrix, the time complexities of EUC, ML-UTSC, SAX-TD, and DTW are $O(n)$, $O(dn)$, $O(wn)$, and $O(n^2)$, respectively.

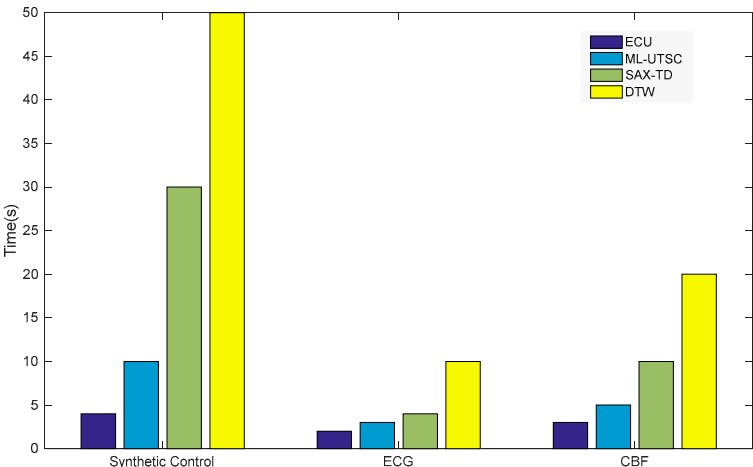

**Figure 8.** Comparison of the efficiency of the four algorithms used.

## 5. Conclusions

In this paper, we proposed a method for combining statistics and metric learning to measure time series similarity. First, the univariate time series data feature was represented by three variables of the mean value, variance, and slope. Next, these variables were used in the metric learning of a three-dimensional Mahalanobis matrix. To obtain a more accurate measurement, the time series was divided into some equal interval segments to obtain the three variable data points with the same weights. Then, the segmented data were used in the Mahalanobis matrix metric learning to ensure a more precise classification. Finally, the classification accuracy, the dimension reduction rate, and the time efficiency were compared with previously reported well-performing methods, including SAX_TD and DTW. In most of the datasets, the classification error rate of our proposed method was lower than SAX_TD and DTW, while the reduction rate and time efficiency were higher.

The PGDM algorithm in metric learning was adopted, which transforms metric learning into a convex optimization problem with constraints. This method makes the time efficiency lower in the learning Mahalanobis matrix. In the future, a deep study on metric learning, such as LMNN and ITML, will be selected to improve efficiency.

**Author Contributions:** For this research, H.W. and N.W. designed the concept of the research; S.K. implemented experimental design; H.W. and K.S. conducted data analysis; K.S. wrote the draft paper; N.W. reviewed and edited the whole paper; N.W. acquired the funding. All authors have read and agreed to the published version of the manuscript.

**Funding:** This work was funded by the National Natural Science Foundation of China under Grant (No. 61772152), and the Basic Research Project (No. JCKY2017604C010, JCKY2019604C004). National key research and development plan (2018YFC0806800); basic technology research project (JSQB2017206C002); pre-research project (10201050201).

**Conflicts of Interest:** The authors declare no conflict of interest.

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
