# Peer review of "A Metric Learning-Based Univariate Time Series Classification Method"

_information, doi:10.3390/info11060288_

Round 1
Reviewer 1 Report
Proposed manuscript is called A Metric Learning-Based Univariate Time Series Classification Method. Manuscript deals with a novel methodology of classification based on Mahalanobis distance.
The manuscript is quite clearly written but the key parts shall be explained much better (I mean chapters 3.3 and 4.2) and the level of English should be improved.
A have only one general critical comment: The manuscript is somewhere between the theoretical research and new application. In the theoretical article I would welcome more novelty, in applied research article I would welcome more experimental results from the real world. Moreover I miss exact comparison to similar classification approaches.
I can see a potential in the manuscript. Owing facts mentioned above I recommend to accept the manuscript after MINOR REVISION.
Reviewer 2 Report
Review of "A Metric Learning-Based Univariate Time Series Classification Method" by Song et al. (2020)
The authors presented an interesting paper addressing the classification and reduction of high-dimension in time series. They proposed Mahalanobis distance to classify univariate time series and reduce dimension.
I think that the paper presented novel method because, they classify univariate time series data by Mahalanobis metric and obtained a metric-based learning with good classification effect. In the aplication, the time series dimensions was reduced (L351-352), being well represented in Figure 7.
I have some comments/suggestions and references to improve the manuscript, in order to accept it for final publication.
Comments/Suggestions:
L25-26: "...the stock market (Drozdz et al., 2007), medical diagnosis (Shah et al., 2017), sensor detection (Verbesselt et al., 2010), and marine
biology (Contreras-Reyes et al., 2016), among others. With..."
L60: "Ye and Keogh [22] and Grabocka et al. [23] presented shapelet..."
L87-89: Fix the presentation of the variables in the text.
L94: Delete "(3)". After Eq. (3): "Then, the equations in (3) yields a linear system easy to solve."
L123: q_t, t=1,...,m; c_s, s=1,...,n,
Eq (7), L129-130: DTW(q_t,c_s), R(q_t,c_s)
L129: Minimum distance is not determined by (7)? What is the difference with (7)?
L136-141: Please, write these lines in a box as a computational algorithm.
L156 (and other parts): use (.,.,.) notation instead <.,.,.> (this last is for inner product).
L157: Delete "(10),(11),(12)", and move "where... point t" to after (12).
L166: "... slope values (colineality problem)".
L215: "a "good" Mahalanobis matrix".
L237: "... optimization, usually the classification performs well."
L241: n? or natural logarithmic?
References:
Drozdz, S., Forczek, M., Kwapien, J., Oswie, P., Rak, R. (2007). Stock market return distributions: From past to present. Physica A 383, 59-64.
Shah, S.M.S., Batool, S., Khan, I., Ashraf, M.U., Abbas, S. H., Hussain, S. A. (2017). Feature extraction through parallel probabilistic principal
component analysis for heart disease diagnosis. Physica A 482, 796-807.
Verbesselt, J., Hyndman, R., Newnham, G., Culvenor, D. (2010). Detecting trend and seasonal changes in satellite image time series. Remote Sens.
Environ. 114, 106–115.
Contreras-Reyes, J.E., Canales, T.M., Rojas, P.M. (2016). Influence of climate variability on anchovy reproductive timing off northern Chile. J. Mar.
Sys. 164, 67-75.
Round 2
Reviewer 2 Report
No further comments. All of my comments have been well addressed by the authors. Good work.